# Debiasing Evidence Approximations: On Importance-Weighted Autoencoders and Jackknife Variational Inference

**Sebastian Nowozin**
Machine Intelligence and Perception
Microsoft Research, Cambridge, UK
`Sebastian.Nowozin@microsoft.com`

## Abstract

The importance-weighted autoencoder (IWAE) approach of Burda et al. (2015) defines a sequence of increasingly tighter bounds on the marginal likelihood of latent variable models. Recently, Cremer et al. (2017) reinterpreted the IWAE bounds as ordinary variational evidence lower bounds (ELBO) applied to increasingly accurate variational distributions. In this work, we provide yet another perspective on the IWAE bounds. We interpret each IWAE bound as a *biased estimator* of the true marginal likelihood where for the bound defined on $K$ samples we show the bias to be of order $O(K^{-1})$. In our theoretical analysis of the IWAE objective we derive asymptotic bias and variance expressions. Based on this analysis we develop *jackknife variational inference* (JVI), a family of bias-reduced estimators reducing the bias to $O(K^{-(m+1)})$ for any given $m < K$ while retaining computational efficiency. Finally, we demonstrate that JVI leads to improved evidence estimates in variational autoencoders. We also report first results on applying JVI to learning variational autoencoders.[1]

## 1 Introduction

*Variational autoencoders* (VAE) are a class of expressive probabilistic deep learning models useful for *generative modeling*, *representation learning*, and *probabilistic regression*. Originally proposed in Kingma & Welling (2013) and Rezende et al. (2014), VAEs consist of a probabilistic model as well as an approximate method for maximum likelihood estimation. In the generative case, the *model* is defined as

$$p(x) = \int p_\theta(x|z) \, p(z) \, \mathrm{d}z, \tag{1}$$

where $z$ is a latent variable, typically a high dimensional vector; the corresponding prior distribution $p(z)$ is fixed and typically defined as a standard multivariate Normal distribution $\mathcal{N}(0, I)$. To achieve an expressive marginal distribution $p(x)$, we define $p_\theta(x|z)$ through a neural network, making the model (1) a deep probabilistic model.

Maximum likelihood estimation of the parameters $\theta$ in (1) is intractable, but Kingma & Welling (2013) and Rezende et al. (2014) propose to instead maximize the *evidence lower-bound* (ELBO),

$$\log p(x) \quad \geq \quad \mathbb{E}_{z \sim q_\omega(z|x)} \left[ \log \frac{p_\theta(x|z) \, p(z)}{q_\omega(z|x)} \right] \tag{2}$$

$$=: \quad \mathcal{L}_E. \tag{3}$$

Here, $q_\omega(z|x)$ is an auxiliary *inference network*, parametrized by $\omega$. Simultaneous optimization of (2) over both $\theta$ and $\omega$ performs approximate maximum likelihood estimation in the model $p(x)$ of (1) and forms the standard VAE estimation method.

---

[1]The implementation is available at `https://github.com/Microsoft/jackknife-variational-inference`

In practice $\mathcal{L}_E$ is estimated using Monte Carlo: we draw $K$ samples $z_i \sim q_\omega(z|x)$, then use the unbiased estimator $\hat{\mathcal{L}}_E$ of $\mathcal{L}_E$,

$$\hat{\mathcal{L}}_E = \frac{1}{K} \sum_{i=1}^{K} \log \frac{p_\theta(x|z_i)\, p(z_i)}{q_\omega(z_i|x)}. \tag{4}$$

The VAE approach is empirically very successful but are there fundamental limitations? One limitation is the quality of the model $p_\theta(x|z)$: this model needs to be expressive enough to model the true distribution over $x$. Another limitation is that $\mathcal{L}_E$ is only a lower-bound to the true likelihood. Is this bound tight? It can be shown, Kingma & Welling (2013), that when $q(z|x) = p(z|x)$ we have $\mathcal{L}_E = \log p(x)$, hence (2) becomes *exact*. Therefore, we should attempt to choose an expressive class of distributions $q(z|x)$ and indeed recent work has extensively investigated richer variational families. We discuss these methods in Section 7 but now review the importance weighted autoencoder (IWAE) method we build upon.

## 2 BURDA'S IMPORTANCE-WEIGHTED AUTOENCODER (IWAE) BOUND

The *importance weighted autoencoder* (IWAE) method Burda et al. (2015) seemingly deviates from (2) in that they propose the IWAE objective, defined for an integer $K \geq 1$,

$$\log p(x) \quad \geq \quad \mathbb{E}_{z_1,\ldots,z_K \sim q_\omega(z|x)} \left[ \log \frac{1}{K} \sum_{i=1}^{K} \frac{p_\theta(x|z_i)\, p(z)}{q_\omega(z_i|x)} \right] \tag{5}$$

$$=: \quad \mathcal{L}_K. \tag{6}$$

We denote with $\hat{\mathcal{L}}_K$ the empirical version which takes one sample $z_1, \ldots, z_K \sim q_\omega(z|x)$ and evaluates the inner expression in (6). We can see that $\mathcal{L}_1 = \mathcal{L}_E$, and indeed Burda et al. (2015) further show that

$$\mathcal{L}_E = \mathcal{L}_1 \leq \mathcal{L}_2 \leq \cdots \leq \log p(x), \tag{7}$$

and $\lim_{K \to \infty} \mathcal{L}_K = \log p(x)$. These results are a strong motivation for the use of $\mathcal{L}_K$ to estimate $\theta$ and the IWAE method can often significantly improve over $\mathcal{L}_E$. The bounds $\mathcal{L}_K$ seem quite different from $\mathcal{L}_E$, but recently Cremer et al. (2017) and Naesseth et al. (2017) showed that an exact correspondence exists: any $\mathcal{L}_K$ can be converted into the standard form $\mathcal{L}_E$ by defining a modified distribution $q_{\text{IW}}(z|x)$ through an importance sampling construction.

We now analyze the IWAE bound $\hat{\mathcal{L}}_K$ in more detail. Independently of our work Rainforth et al. (2017a) has analysed nested Monte Carlo objectives, including the IWAE bound as special case. Their analysis includes results equivalent to our Proposition 1 and 2.

## 3 ANALYSIS OF THE IWAE BOUND

We now analyze the statistical properties of the IWAE estimator of the log-marginal likelihood. Basic consistency results have been shown in Burda et al. (2015); here we provide more precise results and add novel asymptotic results regarding the bias and variance of the IWAE method. Our results are given as expansions in the order $K$ of the IWAE estimator but do involve moments $\mu_i$ which are unknown to us. The jackknife method in the following sections will effectively circumvent the problem of not knowing these moments.

**Proposition 1** (Expectation of $\hat{\mathcal{L}}_K$)**.** *Let $P$ be a distribution supported on the positive real line and let $P$ have finite moments of all order. Let $K \geq 1$ be an integer. Let $w_1, w_2, \ldots, w_K \sim P$ independently. Then we have asymptotically, for $K \to \infty$,*

$$\mathbb{E}[\hat{\mathcal{L}}_K] = \mathbb{E}\left[\log \frac{1}{K} \sum_{i=1}^{K} w_i\right] \quad = \quad \log \mathbb{E}[w] - \frac{1}{K} \frac{\mu_2}{2\mu^2} + \frac{1}{K^2}\left(\frac{\mu_3}{3\mu^3} - \frac{3\mu_2^2}{4\mu^4}\right)$$

$$- \frac{1}{K^3}\left(\frac{\mu_4}{4\mu^4} - \frac{3\mu_2^2}{4\mu^4} - \frac{10\mu_3\mu_2}{5\mu^5}\right) + o(K^{-3}), \tag{8}$$

*where $\mu_i := \mathbb{E}_P[(w - \mathbb{E}_P[w])^i]$ is the $i$'th central moment of $P$ and $\mu := \mathbb{E}_P[w]$ is the mean.*

*Proof.* See Appendix A, page 12. □

The above result directly gives the bias of the IWAE method as follows.

**Corollary 1** (Bias of $\hat{\mathcal{L}}_K$)**.** *If we see $\hat{\mathcal{L}}_K$ as an estimator of $\log p(x)$, then for $K \to \infty$ the bias of $\hat{\mathcal{L}}_K$ is*

$$
\begin{aligned}
\mathbb{B}[\hat{\mathcal{L}}_K] &= \mathbb{E}[\hat{\mathcal{L}}_K] - \log \mathbb{E}[w] & (9) \\
&= -\frac{1}{K}\frac{\mu_2}{2\mu^2} + \frac{1}{K^2}\left(\frac{\mu_3}{3\mu^3} - \frac{3\mu_2^2}{4\mu^4}\right) \\
&\quad -\frac{1}{K^3}\left(\frac{\mu_4}{4\mu^4} - \frac{3\mu_2^2}{4\mu^4} - \frac{10\mu_3\mu_2}{5\mu^5}\right) + o(K^{-3}). & (10)
\end{aligned}
$$

*Proof.* The bias (10) follows directly by subtracting the true value $\log p(x) = \log \mathbb{E}[w]$ from the right hand side of (8). □

The above result shows that the bias is reduced at a rate of $O(1/K)$. This is not surprising because the IWAE estimator is a smooth function applied to a sample mean. The coefficient of the leading $O(1/K)$ bias term uses the ratio $\mu_2/\mu^2$, the variance divided by the squared mean of the $P$ distribution. The quantity $\sqrt{\mu_2/\mu^2}$ is known as the *coefficient of variation* and is a common measure of dispersion of a distribution. Hence, for large $K$ the bias of $\hat{\mathcal{L}}_K$ is small when the coefficient of variation is small; this makes sense because in case the dispersion is small the logarithm function behaves like a linear function and few bias results. The second-order and higher-order terms takes into account higher order properties of $P$.

The bias is the key quantity we aim to reduce, but every estimator is also measured on its variance. We now quantify the variance of the IWAE estimator.

**Proposition 2** (Variance of $\hat{\mathcal{L}}_K$)**.** *For $K \to \infty$, the variance of $\hat{\mathcal{L}}_K$ is given as follows.*

$$
\mathbb{V}[\hat{\mathcal{L}}_K] = \frac{1}{K}\frac{\mu_2}{\mu^2} - \frac{1}{K^2}\left(\frac{\mu_3}{\mu^3} - \frac{5\mu_2^2}{2\mu^4}\right) + o(K^{-2}). \tag{11}
$$

*Proof.* See Appendix A, page 13. □

Both the bias $\mathbb{B}[\hat{\mathcal{L}}_K]$ and the variance $\mathbb{V}[\hat{\mathcal{L}}_K]$ vanish for $K \to \infty$ at a rate of $O(1/K)$ with similar coefficients. This leads to the following result which was already proven in Burda et al. (2015).

**Corollary 2** (Consistency of $\hat{\mathcal{L}}_K$)**.** *For $K \to \infty$ the estimator $\hat{\mathcal{L}}_K$ is* consistent, *that is, for all $\epsilon > 0$*

$$
\lim_{K \to \infty} P(|\hat{\mathcal{L}}_K - \log p(x)| \geq \epsilon) = 0. \tag{12}
$$

*Proof.* See Appendix A, page 13. □

How good are the asymptotic results? This is hard to say in general because it depends on the particular distribution $P(w)$ of the weights. In Figure 1 we show both a simple and challenging case to demonstrate the accuracy of the asymptotics.

The above results are reassuring evidence for the IWAE method, however, they cannot be directly applied in practice because we do not know the moments $\mu_i$. One approach is to estimate the moments from data, and this is in fact what the delta method variational inference (DVI) method does, Teh et al. (2007), (see Appendix B, page 14); however, estimating moments accurately is difficult. We avoid the difficulty of estimating moments by use of the *jackknife*, a classic debiasing method. We now review this method.

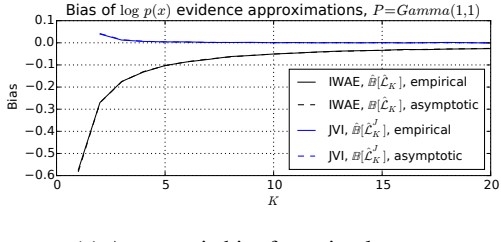
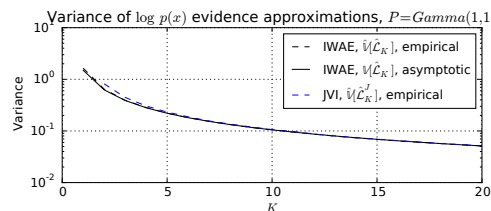

(a) Asymptotic bias for a simple case.

(b) Asymptotic variance for a simple case.

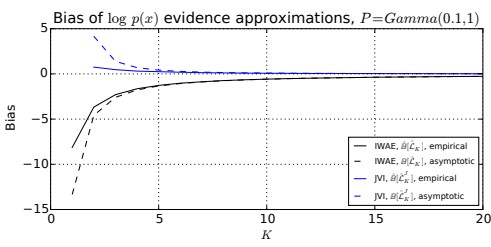
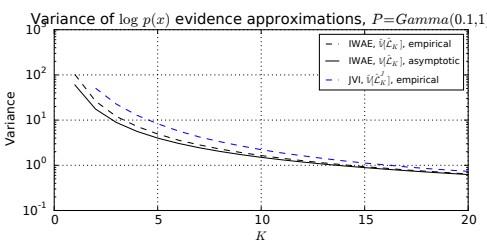

(c) Asymptotic bias for a challenging case.

(d) Asymptotic variance for a challenging case.

Figure 1: Comparing asymptotics with empirical values of bias and variance on $P = \text{Gamma}(1, 1)$ using 100,000 independent evaluations: (a)-(b) shows a simple case, $P = \text{Gamma}(1, 1)$, and (c)-(d) shows a challenging case. Observation: (a) the IWAE is negatively biased, underestimating $\log p(x)$, with asymptotic expression (10) agreeing very well with empirical bias; (b) empirical and asymptotic variance (11) in good agreement. (c) for a challenging case, the bias asymptotics match empirical estimates for $K \geq 10$; (d) in the challenging case, the variance asymptotics match empirical estimates for $K \geq 10$.

## 4 A BRIEF REVIEW OF THE JACKKNIFE

We now provide a brief review of the *jackknife* and *generalized jackknife* methodology. Our presentation deviates from standard textbook introductions, Miller (1974), in that we also review higher-order variants.

The *jackknife* methodology is a classic resampling technique originating with Quenouille (1949; 1956) in the 1950s. It is a generally applicable technique for estimating the *bias* $\mathbb{B}[\hat{T}] = \mathbb{E}[\hat{T}] - T$ and the *variance* $\mathbb{V}[\hat{T}]$ of an estimator $\hat{T}$. Our focus is on estimating and correcting for bias.

The basic intuition is as follows: in many cases it is possible to write the expectation of a *consistent* estimator $\hat{T}_n$ evaluated on $n$ samples as an asymptotic expansion in the sample size $n$, that is, for large $n \to \infty$ we have

$$\mathbb{E}[\hat{T}_n] = T + \frac{a_1}{n} + \frac{a_2}{n^2} + \dots. \tag{13}$$

In particular, this is possible in case the estimator is consistent and a smooth function of linear statistics. If an expansion (13) is possible, then we can take a linear combination of two estimators $\hat{T}_n$ and $\hat{T}_{n-1}$ to cancel the first order term,

$$
\begin{aligned}
\mathbb{E}[n\hat{T}_n - (n-1)\hat{T}_{n-1}] &= n\left(T + \frac{a_1}{n} + \frac{a_2}{n^2}\right) - (n-1)\left(T + \frac{a_1}{n-1} + \frac{a_2}{(n-1)^2}\right) + O(n^{-2}) \\
&= T + \frac{a_2}{n} - \frac{a_2}{n-1} + O(n^{-2}) \tag{14} \\
&= T - \frac{a_2}{n(n-1)} + O(n^{-2}) \tag{15} \\
&= T + O(n^{-2}). \tag{16}
\end{aligned}
$$

Therefore, the *jackknife bias-corrected estimator* $\hat{T}_J := n\hat{T}_n - (n-1)\hat{T}_{n-1}$ achieves a reduced bias of $O(n^{-2})$. For $\hat{T}_{n-1}$ any estimator which preserves the expectation (13) can be used. In practice

we use the original sample of size $n$ to create $n$ subsets of size $n-1$ by removing each individual sample once. Then, the empirical average of $n$ estimates $\hat{T}_{n-1}^{\backslash i}$, $i = 1, \ldots, n$ is used in place of $\hat{T}_{n-1}$. In Sharot (1976) this construction was proved optimal in terms of maximally reducing the variance of $\hat{T}_J$ for any given sample size $n$.

In principle, the above bias reduction (16) can be repeated to further reduce the bias to $O(n^{-3})$ and beyond. The possibility of this was already hinted at in Quenouille (1956) by means of an example.[2] A fully general and satisfactory solution to higher-order bias removal was only achieved by the *generalized jackknife* of Schucany et al. (1971), considering estimators $\hat{T}_G$ of order $m$, each having the form,

$$\hat{T}_G^{(m)} = \sum_{j=0}^{m} c(n, m, j)\, \hat{T}_{n-j}. \tag{17}$$

The form of the coefficients $c(n, m, j)$ in (17) are defined by the ratio of determinants of certain Vandermonde matrices, see Schucany et al. (1971). In a little known result, an analytic solution for $c(n, m, j)$ is given by Sharot (1976). We call this form the *Sharot coefficients*, (Sharot, 1976, Equation (2.5) with $r = 1$), defined for $m < n$ and $0 \le j \le m$,

$$c(n, m, j) = (-1)^j \frac{(n-j)^m}{(m-j)!\, j!}. \tag{18}$$

The generalized jackknife estimator $\hat{T}_G^{(m)}$ achieves a bias of order $O(m^{-(j+1)})$, see Schucany et al. (1971). For example, the classic jackknife is recovered because $c(n, 1, 0) = n$ and $c(n, 1, 1) = -(n-1)$. As an example of the second-order generalized jackknife we have

$$c(n, 2, 0) = \frac{n^2}{2}, \qquad c(n, 2, 1) = -(n-1)^2, \qquad c(n, 2, 2) = \frac{(n-2)^2}{2}. \tag{19}$$

The *variance* of generalized jackknife estimators is more difficult to characterize and may in general decrease or increase compared to $\hat{T}_n$. Typically we have $\mathbb{V}[\hat{T}_G^{(m+1)}] > \mathbb{V}[\hat{T}_G^{(m)}]$ with asymptotic rates being the same.

The generalized jackknife is not the only method for debiasing estimators systematically. One classic method is the *delta method for bias correction* Small (2010). Two general methods for debiasing are the *iterated bootstrap for bias correction* (Hall, 2016, page 29) and the *debiasing lemma* McLeish (2010); Strathmann et al. (2015); Rhee & Glynn (2015). Remarkably, the debiasing lemma exactly debiases a large class of estimators.

The delta method bias correction has been applied to variational inference by Teh et al. (2007); we provide novel theoretical results for the method in Appendix B, page 14.

## 5 JACKKNIFE VARIATIONAL INFERENCE (JVI)

We now propose to apply the generalized jackknife for bias correction to variational inference by debiasing the IWAE estimator. The resulting estimator of the log-marginal likelihood will have significantly reduced bias, however, in contrast to the ELBO and IWAE, *it is no longer a lower bound* on the true log-marginal likelihood. Moreover, it can have increased variance compared to both IWAE and ELBO estimators. We will empirically demonstrate that the variance is comparable to the IWAE estimate and that the bias reduction is very effective in improving our estimates.

**Definition 1** (Jackknife Variational Inference (JVI)). *Let $K \geq 1$ and $m < K$. The jackknife variational inference estimator of the evidence of order $m$ with $K$ samples is*

$$\hat{\mathcal{L}}_K^{J,m} \quad := \quad \sum_{j=0}^{m} c(K, m, j)\, \bar{\mathcal{L}}_{K-j}, \tag{20}$$

*where $\bar{\mathcal{L}}_{K-j}$ is the empirical average of one or more IWAE estimates obtained from a subsample of size $K - j$, and $c(K, m, j)$ are the* Sharot coefficients *defined in (18). In this paper we use all*

---

[2]Which was subtly wrong and did not reduce the bias to $O(n^{-2})$ as claimed, see Schucany et al. (1971).

possible $\binom{K}{K-j}$ subsets, that is,

$$\bar{\mathcal{L}}_{K-j} := \frac{1}{\binom{K}{K-j}} \sum_{i=1}^{\binom{K}{K-j}} \hat{\mathcal{L}}_{K-j}(Z_i^{(K-j)}), \tag{21}$$

where $Z_i^{(K-j)}$ is the $i$'th subset of size $K - j$ among all $\binom{K}{K-j}$ subsets from the original samples $Z = (z_1, z_2, \ldots, z_K)$. We further define $\mathcal{L}_K^{J,m} = \mathbb{E}_Z[\hat{\mathcal{L}}_K^{J,m}]$.

From the above definition we can see that JVI strictly generalizes the IWAE bound and therefore also includes the standard ELBO objective: we have the IWAE case for $\hat{\mathcal{L}}_K^{J,0} = \hat{\mathcal{L}}_K$, and the ELBO case for $\hat{\mathcal{L}}_1^{J,0} = \hat{\mathcal{L}}_E$.

## 5.1 Analysis of $\hat{\mathcal{L}}_K^{J,m}$

The proposed family of JVI estimators has less bias than the IWAE estimator. The following result is a consequence of the existing theory on the generalized jackknife bias correction.

**Proposition 3** (Bias of $\hat{\mathcal{L}}_K^{J,m}$). *For any $K \geq 1$ and $m < K$ we have that the bias of the JVI estimate satisfies*

$$\mathbb{B}[\hat{\mathcal{L}}_K^{J,m}] = \mathbb{E}[\hat{\mathcal{L}}_K^{J,m} - \log p(x)] = \mathcal{L}_K^{J,m} - \log p(x) = O(K^{-(m+1)}). \tag{22}$$

*Proof.* The JVI estimator $\hat{\mathcal{L}}_K^{J,m}$ is the application of the higher-order jackknife to the IWAE estimator which has an asymptotic expansion of the bias (10) in terms of orders of $1/K$. The stated result is then a special case of (Schucany et al., 1971, Theorem 4.2). $\quad\square$

We show an illustration of higher-order bias removal in Appendix C, page 15. It is more difficult to characterize the variance of $\hat{\mathcal{L}}_K^{J,m}$. Empirically we observe that $\mathbb{V}[\hat{\mathcal{L}}_K^{J,m}] < \mathbb{V}[\hat{\mathcal{L}}_K^{J,m'}]$ for $m < m'$, but we have been unable to derive a formal result to this end. Note that the variance is over the sampling distribution of $q(z|x)$, so we can always reduce the variance by averaging multiple estimates $\hat{\mathcal{L}}_K^{J,m}$, whereas we cannot reduce bias this way. Therefore, reducing bias while increasing variance is a sensible tradeoff in our application.

## 5.2 Efficient Computation of $\hat{\mathcal{L}}_K^{J,m}$

We now discuss how to efficiently compute (20). For typical applications, for example in variational autoencoders, we will use small values of $K$, say $K < 100$. However, even with $K = 50$ and $m = 2$ there are already 1276 IWAE estimates to compute in (20–21). Therefore efficient computation is important to consider. One property that helps us is that all these IWAE estimates are related because they are based on subsets of the *same* weights. The other property that is helpful is that computation of the $K$ weights is typically orders of magnitude more expensive than elementary summation operations required for computation of (21).

We now give a general algorithm for computing the JVI estimator $\hat{\mathcal{L}}_K^{J,m}$, then give details for efficient implementation on modern GPUs and state complexity results.

Algorithm 1 computes log-weights and implements equations (20–21) in a numerically robust manner.[3]

**Proposition 4** (Complexity of Algorithm 1). *Given $K \geq 1$ and $m \leq K/2$ the complexity of Algorithm 1 is*

$$O\left(Ke^m \left(\frac{K}{m}\right)^m\right). \tag{23}$$

*Proof.* See Appendix C, page 15. $\quad\square$

---

[3]As usual, the log-sum-exp operation needs to be numerically robustly implemented.

---

**Algorithm 1** Computing $\hat{\mathcal{L}}_K^{J,m}$, the jackknife variational inference estimator

---

1: **function** COMPUTEJVI($m, K, p, q, x$)
2:     **for** $i = 1, \ldots, K$ **do**
3:         Sample $z_i \sim q(z|x)$
4:         $v_i \leftarrow \log p(x|z_i) + \log p(z_i) - \log q(z_i|x)$
5:     **end for**
6:     $L \leftarrow 0$
7:     **for** $j = 0, \ldots, m$ **do**
8:         $\bar{L} \leftarrow 0$                                                      $\triangleright \bar{\mathcal{L}}_{K-j}$
9:         **for** $S \in \text{EnumerateSubsets}(\{1, \ldots, K\}, K - j)$ **do**     $\triangleright$ list all subsets of size $K - j$
10:           $\bar{L} \leftarrow \bar{L} + \log \sum_{s \in S} \exp v_s - \log(K - j)$           $\triangleright$ IWAE estimate for subset $S$
11:         **end for**
12:         $L \leftarrow L + \frac{c(K,m,j)}{\binom{K}{K-j}} \bar{L}$                                  $\triangleright$ Using equation (18)
13:     **end for**
14:     **return** $L$                                             $\triangleright$ JVI estimate $\hat{\mathcal{L}}_K^{J,m}$
15: **end function**

---

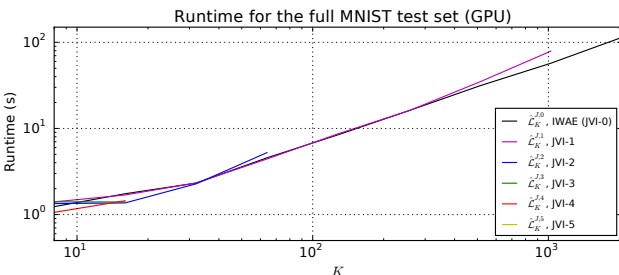

Figure 2: Runtime evaluation of the $\hat{\mathcal{L}}_K^{J,m}$ estimators.

The above algorithm is suitable for CPU implementation; to utilize modern GPU hardware efficiently we can instead represent the second part of the algorithm using matrix operations. We provide further details in Appendix C, page 16. Figure 2 demonstrates experimental runtime evaluation on the MNIST test set for different JVI estimators. We show all JVI estimators with less than 5,000 total summation terms. The result demonstrates that runtime is largely independent of the order of the JVI correction and only depends linearly on $K$.

### 5.3   VARIATIONS OF THE JVI ESTIMATOR

Variations of the JVI estimator with improved runtime exist. Such reduction in runtime are possible if we consider evaluating only a fraction of all possible subsets in (21). When tractable, our choice of evaluating all subsets is generally preferable in terms of variance of the resulting estimator. However, to show that we can even reduce bias to order $O(K^{-K})$ at cost $O(K)$ we consider the estimator

$$\hat{\mathcal{L}}_K^X \;\; := \;\; \sum_{j=0}^{K-1} c(K, K-1, j) \, \hat{\mathcal{L}}_{K-j}(Z_{1:(K-j)}) \tag{24}$$

$$= \;\; c(K, K-1, K-1) \, \log(\exp(v_K)) \tag{25}$$

$$+ c(K, K-1, K-2) \, \log\left(\frac{1}{2}(\exp(v_{K-1}) + \exp(v_K))\right) \tag{26}$$

$$+ \cdots + c(K, K-1, 0) \, \log\left(\frac{1}{K} \sum_{i=1}^{K} \exp(v_i)\right). \tag{27}$$

The sum (25–27) can be computed in time $O(K)$ by keeping a running partial sum $\sum_{i=1}^{k} \exp(v_i)$ for $k \leq K$ and by incrementally updating this sum[4], meaning that (24) can be computed in $O(K)$

---

[4]To do this in a numerically stable manner, we need to use streaming log-sum-exp computations, see for example `http://www.nowozin.net/sebastian/blog/streaming-log-sum-exp-computation.html`

overall. As a generalized jackknife estimate $\hat{\mathcal{L}}_K^X$ has bias $O(K^{-K})$. We do not recommend its use in practice because its variance is large, however, developing estimators between the two extremes of taking one set and taking all sets of subsets of a certain size seems a good way to achieve high-order bias reduction while controlling variance.

# 6 EXPERIMENTS

We now empirically validate our key claims regarding the JVI method: 1. JVI produces better estimates of the marginal likelihood by reducing bias, even for small $K$; and 2. Higher-order bias reduction is more effective than lower-order bias reduction;

To this end we will use variational autoencoders trained on MNIST. Our setup is purposely identical to the setup of Tomczak & Welling (2016), where we use the dynamically binarized MNIST data set of Salakhutdinov & Murray (2008). Our numbers are therefore directly comparable to the numbers reported in the above works. Our implementation is available at `https://github.com/Microsoft/jackknife-variational-inference`.

We first evaluate the accuracy of evidence estimates given a fixed model. This setting is useful for assessing model performance and for model comparison.

## 6.1 JVI AS EVALUATION METHOD

We train a regular VAE on the dynamically binarized MNIST dataset using either the ELBO, IWAE, or JVI-1 objective functions. We use the same two-layer neural network architecture with 300 hidden units per layer as in (Tomczak & Welling, 2016). We train on the first 50,000 training images, using 10,000 images for validation. We train with SGD for 5,000 epochs and take as the final model the model with the maximum validation objective, evaluated after every training epoch. Hyperparameters are the batch size in $\{1024, 4096\}$ and the SGD step size in $\{0.1, 0.05, 0.01, 0.005, 0.001\}$. The final model achieving the best validation score is evaluated once on the MNIST test set. All our models are implemented using *Chainer* (Tokui et al., 2015) and run on a NVidia Titan X.

For three separate models, trained using the ordinary ELBO, IWAE, and JVI-1 objectives, we then estimate the marginal log-likelihood (evidence) on the MNIST test set. For evaluation we use JVI estimators up to order five in order to demonstrate higher-order bias reduction. Among all possible JVI estimators up to order five we evaluate only those JVI estimators whose total sum of IWAE estimates has less than 5,000 terms. For example, we do not evaluate $\hat{\mathcal{L}}_{32}^{J,3}$ because it contains $\binom{32}{0} + \binom{32}{1} + \binom{32}{2} + \binom{32}{3} = 5489$ terms.[5]

Figure 3 shows the evidence estimates for three models. We make the following observations, applying to all plots: 1. Noting the logarithmic x-axis we can see that higher-order JVI estimates are more than one order of magnitude more accurate than IWAE estimates. 2. The quality of the evidence estimates empirically improves monotonically with the order of the JVI estimator; 3. In absolute terms the improvements in evidence estimates is larges for small values of $K$, which is what is typically used in practice; 4. The higher-order JVI estimators remove low-order bias but significant higher-order bias remains even for $K = 64$, showing that on real VAE log-weights the contribution of higher-order bias to the evidence error is large; 5. The standard error of each test set marginal likelihood (shown as error bars, best visible in a zoomed version of the plot) is comparable across all JVI estimates; this empirically shows that higher-order bias reduction does not lead to high variance.

## 6.2 JVI AS A TRAINING OBJECTIVE

We now report preliminary results on learning models using the JVI objectives. The setting is the same as in Section 6.1 and we report the average performance of five independent runs.

Table 1 reports the results. We make the following observations: 1. When training on the IWAE and JVI-1 objectives, the respective score by the ELBO objective is impoverished and this effect makes

---

[5]We do this because we discovered numerical issues for large sums of varying size and found all summations of less than a few thousand terms not to have this problem but we are looking into a way to compute more summation terms in a fast and robust manner.

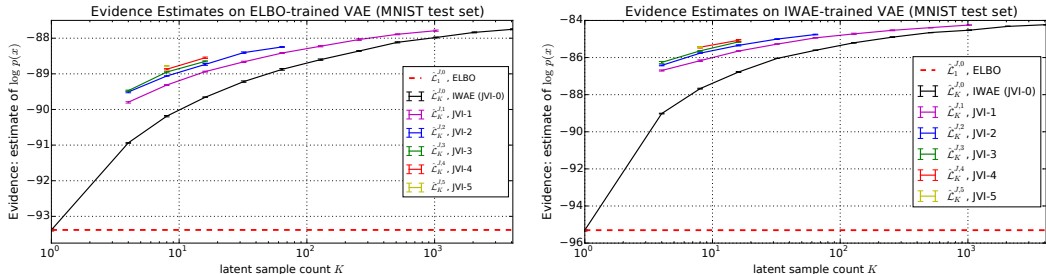

(a) Evidence estimates on VAE-trained MNIST model. (b) Evidence estimates on IWAE-trained MNIST model.

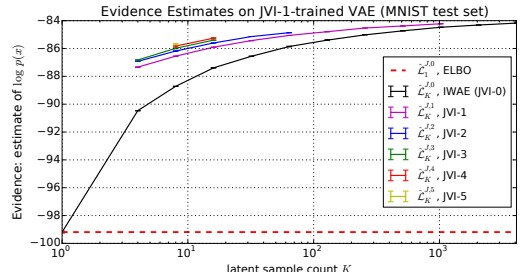

(c) Evidence estimates on JVI-1-trained MNIST model.

Figure 3: Comparing evidence approximations on MNIST variational autoencoders: (a) VAE trained using the ELBO objective $\hat{\mathcal{L}}_E$; (b) VAE trained using the IWAE objective $\hat{\mathcal{L}}_K$ with $K = 32$; (c) VAE trained using the JVI-1 objective $\hat{\mathcal{L}}_K^{J,1}$ with $K = 32$.

| Training objective ($K = 32$) | Evaluation objective (nats), $K = 32$ | | | |
|---|---|---|---|---|
| | ELBO | IWAE | JVI-1 | JVI-2 |
| ELBO | $-93.38 \pm 0.03$ | $-89.22 \pm 0.02$ | $-88.66 \pm 0.02$ | $-88.40 \pm 0.02$ |
| IWAE | $-95.30 \pm 0.05$ | $-86.05 \pm 0.01$ | $-85.28 \pm 0.03$ | $-85.01 \pm 0.02$ |
| JVI-1 | $-99.19 \pm 0.06$ | $-86.56 \pm 0.02$ | $-85.43 \pm 0.02$ | $-85.14 \pm 0.01$ |

Table 1: Evaluating models trained using ELBO, IWAE, and JVI-1 learning objectives.

sense in light of the work of Cremer et al. (2017). Interestingly the effect is stronger for JVI-1. 2. The model trained using the JVI-1 objective falls slightly behind the IWAE model, which is surprising because the evidence is clearly better approximated as demonstrated in Section 6.1. We are not sure what causes this issue, but have two hypotheses: *First*, in line with recent findings (Rainforth et al., 2017b) a tighter log-evidence approximation could lead to poor encoder models. In such case it is worth exploring two separate learning objectives for the encoder and decoder; for example, using an ELBO for training the encoder, and an IWAE or JVI-1 objective for training the decoder. *Second*, because JVI estimators are no longer bounds it could be the case that during optimization of the learning objective a decoder is systematically learned in order to amplify positive bias in the log-evidence.

# 7   RELATED WORK

The IWAE bound and other Monte Carlo objectives have been analyzed by independently by Rainforth et al. (2017a). Their analysis is more general than our IWAE analysis, but does not propose a method to reduce bias.

Delta-method variational inference (DVI) proposed by Teh et al. (2007) is the closest method we are aware of and we discuss it in detail as well as provide novel results in Appendix B, page 14. Another exciting recent work is perturbative variational inference (Bamler et al., 2017) which considers different objective functions for variational inference; we are not sure whether there exists a deeper relationship to debiasing schemes.

There also exists a large body of work that uses the ELBO objective but considers ways to enlarge the variational family. This is useful because the larger the variational family, the smaller the bias.

Non-linear but invertible transformations of reference densities have been used initially for density estimation in NICE (Dinh et al., 2014) and for variational inference in Hamiltonian variational inference (Salimans et al., 2015). Around the same time the general framework of *normalizing flows* (Rezende & Mohamed, 2015) unified the previous works as some invertible continuous transformation of a distribution. Since then a large number of specialized flows with different computational requirements and flexibility have been constructed: inverse autoregressive flows (Kingma et al., 2016), masked autoregressive flows Papamakarios et al. (2017), and Householder flows (Tomczak & Welling, 2016).

Another way to improve the flexibility of the variational family has been to use implicit models (Mohamed & Lakshminarayanan, 2016) for variational inference; this line of work includes adversarial variational Bayes (Mescheder et al., 2017), wild variational inference (Li & Liu, 2016), deep implicit models (Tran et al., 2017), implicit variational models (Huszár, 2017), and adversarial message passing approximations (Karaletsos, 2016).

## 8 CONCLUSION

In summary we proposed to leverage classic higher-order bias removal schemes for evidence estimation. Our approach is simple to implement, computationally efficient, and clearly improves over existing evidence approximations based on variational inference. More generally our jackknife variational inference debiasing formula can also be used to debias log-evidence estimates coming from annealed importance sampling.

However, one surprising finding from our work is that using our debiased estimates for training VAE models did not improve over the IWAE training objective and this is surprising because apriori a better evidence estimate should allow for improved model learning.

One possible extension to our work is to study the use of other resampling methods for bias reduction; promising candidates are the iterated bootstrap, the Bayesian bootstrap, and the debiasing lemma. These methods could offer further improvements on bias reduction or reduced variance, however, the key challenge is to overcome computational requirements of these methods or, alternatively, to derive key quantities analytically.[6] Application of the debiasing lemma in particular requires the careful construction of a truncation distribution and often produces estimators of high variance.

While variance reduction plays a key role in certain areas of machine learning, our hope is that our work shows that bias reduction techniques are also widely applicable.

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

## APPENDIX A: PROOFS FOR THE IWAE ANALYSIS

### EXPECTATION AND BIAS OF $\hat{\mathcal{L}}_K$

*Proof.* (Of Proposition 1, page 2) To show (8) we apply the *delta method for moments* (Small, 2010, Section 4.3). First, we define the random variable $Y_K := \frac{1}{K} \sum_{i=1}^{K} w_i$ corresponding to the *sample mean* of $w_1, \ldots, w_K$. Because of linearity of expectation we have $\mathbb{E}[Y_K] = \mathbb{E}[w]$. We expand the logarithm function $\log Y_K = \log(\mathbb{E}[w] + (Y_K - \mathbb{E}[w]))$ around $\mathbb{E}[w]$ to obtain

$$\log(\mathbb{E}[w] + (Y_K - \mathbb{E}[w])) \quad = \quad \log \mathbb{E}[w] - \sum_{j=1}^{\infty} \frac{(-1)^j}{j\, \mathbb{E}[w]^j} (Y_K - \mathbb{E}[w])^j. \tag{28}$$

Note that only $Y_K$ is random in (28), all other quantities are constant. Therefore, by taking the expectation on the left and right side of (28) we obtain

$$\mathbb{E}[\log Y_K] \quad = \quad \log \mathbb{E}[w] - \sum_{j=1}^{\infty} \frac{(-1)^j}{j\, \mathbb{E}[w]^j} \mathbb{E}[(Y_K - \mathbb{E}[w])^j]. \tag{29}$$

The right hand side of (29) is expressed in terms of the central moments for $i \geq 2$, $\gamma_i := \mathbb{E}[(Y_K - \mathbb{E}[Y_K])^i]$ of $Y_K$, whereas we are interested in an expression using the central moments $i \geq 2$, $\mu_i := \mathbb{E}[(w - \mathbb{E}[w])^i]$ of $P$. With $\gamma = \mu = \mathbb{E}[w]$ we denote the shared first non-central moment. Because $Y_K$ is a *sample mean* we can use existing results that relate $\gamma_i$ to $\mu_i$. In particular (Angelova, 2012, Theorem 1) gives the relations

$$\gamma \quad = \quad \mu \tag{30}$$

$$\gamma_2 \quad = \quad \frac{\mu_2}{K} \tag{31}$$

$$\gamma_3 \quad = \quad \frac{\mu_3}{K^2} \tag{32}$$

$$\gamma_4 \quad = \quad \frac{3}{K^2}\mu_2^2 + \frac{1}{K^3}\left(\mu_4 - 3\mu_2^2\right) \tag{33}$$

$$\gamma_5 \quad = \quad \frac{10}{K^3}\mu_3\mu_2 + \frac{1}{K^4}\left(\mu_5 - 10\mu_3\mu_2\right). \tag{34}$$

Expanding (29) to order five and using the relations (30) to (34) gives

$$\mathbb{E}[\log Y_K] = \log \mathbb{E}[w] - \frac{1}{2\mu^2}\frac{\mu_2}{K} + \frac{1}{3\mu^3}\frac{\mu_3}{K^2} - \frac{1}{4\mu^4}\left(\frac{3}{K^2}\mu_2^2 + \frac{1}{K^3}\left(\mu_4 - 3\mu_2^2\right)\right)$$

$$+ \frac{1}{5\mu^5}\left(\frac{10}{K^3}\mu_3\mu_2 + \frac{1}{K^4}\left(\mu_5 - 10\mu_3\mu_2\right)\right) + o(K^{-3}). \tag{35}$$

Regrouping the terms by order of $K$ produces the result (8). $\qquad\square$

VARIANCE OF $\hat{\mathcal{L}}_K$

*Proof.* (Of Proposition 2, page 3) We use the definition of the variance and the series expansion of the logarithm function, obtaining

$$\mathbb{V}[\log Y_K] = \mathbb{E}[(\log Y_K - \mathbb{E}[\log Y_K])^2] \tag{36}$$

$$= \mathbb{E}\left[\left(\log\mu - \sum_{i=1}^{\infty}\frac{(-1)^i}{i\mu^i}(Y_K - \mu)^i - \log\mu + \sum_{i=1}^{\infty}\frac{(-1)^i}{i\mu^i}\mathbb{E}[(Y_K - \mu)^i]\right)^2\right]$$

$$= \mathbb{E}\left[\left(\sum_{i=1}^{\infty}\frac{(-1)^i}{i\mu^i}\left(\mathbb{E}[(Y_K - \mu)^i] - (Y_K - \mu)^i\right)\right)^2\right]. \tag{37}$$

By expanding (37) to third order and expanding all products we obtain a moment expansion of $Y_K$ as follows.

$$\mathbb{V}[\log Y_K] \approx \frac{\gamma_2}{\mu^2} - \frac{1}{\mu^3}(\gamma_3 - \gamma_1\gamma_2) + \frac{2}{3\mu^4}(\gamma_4 - \gamma_1\gamma_3) + \frac{1}{4\mu^4}(\gamma_4 - \gamma_2^2) \tag{38}$$

$$- \frac{1}{3\mu^5}(\gamma_5 - \gamma_2\gamma_3) + \frac{1}{9\mu^6}(\gamma_6 - \gamma_3^2). \tag{39}$$

By substituting the sample moments $\gamma_i$ of $Y_K$ with the central moments $\mu_i$ of the original distribution $P$ and simplifying we obtain

$$\mathbb{V}[\log Y_K] = \frac{1}{K}\frac{\mu_2}{\mu^2} - \frac{1}{K^2}\left(\frac{\mu_3}{\mu^3} - \frac{5\mu_2^2}{2\mu^4}\right) + o(K^{-2}). \tag{40}$$

$$\square$$

CONSISTENCY OF $\hat{\mathcal{L}}_K$

*Proof.* We have

$$P(|\hat{\mathcal{L}}_K - \log p(x)| \geq \epsilon) = P(|\hat{\mathcal{L}}_K - \mathbb{E}[\hat{\mathcal{L}}_K] + \mathbb{E}[\hat{\mathcal{L}}_K] - \log p(x)| \geq \epsilon) \tag{41}$$

$$\leq P(|\hat{\mathcal{L}}_K - \mathbb{E}[\hat{\mathcal{L}}_K]| + |\mathbb{E}[\hat{\mathcal{L}}_K] - \log p(x)| \geq \epsilon) \tag{42}$$

$$\leq P(|\hat{\mathcal{L}}_K - \mathbb{E}[\hat{\mathcal{L}}_K]| \geq \frac{\epsilon}{2}) + P(|\mathbb{E}[\hat{\mathcal{L}}_K] - \log p(x)| \geq \frac{\epsilon}{2}). \tag{43}$$

The second term in (43) does not involve a random variable therefore is either zero or one. For large enough $K$ it will always be zero due to (10).

For the first term in (43) we apply Chebyshev's inequality. We set $\tau = \frac{\epsilon}{2\sqrt{\mathbb{V}[\hat{\mathcal{L}}_K]}}$ and have

$$P(|\hat{\mathcal{L}}_K - \mathbb{E}[\hat{\mathcal{L}}_K]| \geq \frac{\epsilon}{2}) = P(|\hat{\mathcal{L}}_K - \mathbb{E}[\hat{\mathcal{L}}_K]| \geq \tau\sqrt{\mathbb{V}[\hat{\mathcal{L}}_K]}) \tag{44}$$

$$\leq \frac{1}{\tau^2} \tag{45}$$

$$= 4\frac{\mathbb{V}[\hat{\mathcal{L}}_K]}{\epsilon^2} \tag{46}$$

$$= O(1/K). \tag{47}$$

Thus, for $K \to \infty$ and any $\epsilon > 0$ we have that (43) has a limit of zero. This establishes convergence in probability and hence consistency. $\qquad\square$

## APPENDIX B: ANALYSIS OF DELTA-METHOD VARIATIONAL INFERENCE (DVI)

**Definition 2** (Delta method Variational Inference (DVI), (Teh et al., 2007))**.**

$$\mathcal{L}_K^D \quad := \quad \mathbb{E}_{z_1,\ldots,z_K \sim q_\omega(z|x)} \left[ \log \frac{1}{K} \sum_{i=1}^{K} w_i + \frac{\hat{w}_2}{2K\hat{w}} \right], \tag{48}$$

*where*

$$w_i \quad = \quad \frac{p(x|z_i)\,p(z_i)}{q_\omega(z_i|w)}, \qquad i = 1, \ldots, K, \tag{49}$$

$$\hat{w}_2 \quad := \quad \frac{1}{K-1} \sum_{i=1}^{K} (w_i - \hat{w})^2, \tag{50}$$

$$\hat{w} \quad := \quad \frac{1}{K} \sum_{i=1}^{K} w_i, \tag{51}$$

*so that $\hat{w}_2$ corresponds to the* sample variance *and $\hat{w}$ corresponds to the sample mean.*

*The practical Monte Carlo estimator of (48) is defined as follows.*

$$z_i \quad \sim \quad q_\omega(z|x), \qquad i = 1, \ldots, K, \tag{52}$$

$$\hat{\mathcal{L}}_K^D \quad := \quad \log \frac{1}{K} \sum_{i=1}^{K} w_i + \frac{\hat{w}_2}{2K\hat{w}}. \tag{53}$$

ANALYSIS OF DELTA METHOD VARIATIONAL INFERENCE

**Proposition 5** (Bias of $\hat{\mathcal{L}}_K^D$)**.** *We evaluate the bias of $\hat{\mathcal{L}}_K^D$ in (53) as follows.*

$$\mathbb{B}[\hat{\mathcal{L}}_K^D] \quad = \quad -\frac{1}{K^2} \left( \frac{\mu_3}{\mu^3} - \frac{3\mu_2^2}{2\mu^4} \right) + o(K^{-2}). \tag{54}$$

*Proof.* Consider the function $f(x,y) = \frac{x}{y^2}$ and its second order Taylor expansion around $(x,y) = (\mu_2, \mu)$,

$$f(\mu_2 + (\hat{\mu}_2 - \mu_2), \mu + (\hat{\mu} - \mu)) \quad \approx \quad \frac{\mu_2}{\mu^2} + \frac{1}{\mu^2}(\hat{\mu}_2 - \mu_2) - \frac{2\mu_2}{\mu^3}(\hat{\mu} - \mu) \tag{55}$$

$$-\frac{2}{\mu^3}(\hat{\mu}_2 - \mu_2)(\hat{\mu} - \mu) + \frac{6\mu_2}{2\mu^4}(\hat{\mu} - \mu)^2. \tag{56}$$

Taking expectations on both sides cancels all linear terms and yields

$$\mathbb{E} \left[ \frac{\hat{\mu}_2}{\hat{\mu}^2} \right] \quad \approx \quad \frac{\mu_2}{\mu^2} - \frac{2}{\mu^3} \mathbb{E}[(\hat{\mu}_2 - \mu_2)(\hat{\mu} - \mu)] + \frac{3\mu_2}{\mu^4} \mathbb{E}[(\hat{\mu} - \mu)^2]. \tag{57}$$

By classic results we have that the expected variance of the sample mean around the true mean is related to the variance by $\mathbb{E}[(\hat{\mu} - \mu)^2] = \mu_2/K$. Furthermore, Zhang (2007) showed a beautiful result about the covariance of sample mean and sample variance for arbitrary random variables, namely that

$$\text{Cov}[\hat{\mu}_2, \hat{\mu}] = \mathbb{E}[(\hat{\mu}_2 - \mu_2)(\hat{\mu} - \mu)] = \mu_3/K. \tag{58}$$

Using both results in (57) produces

$$\mathbb{E} \left[ \frac{\hat{\mu}_2}{\hat{\mu}^2} \right] \quad = \quad \frac{\mu_2}{\mu^2} - \frac{1}{K} \left( \frac{2\mu_3}{\mu^3} - \frac{3\mu_2^2}{\mu^4} \right) + o(K^{-1}). \tag{59}$$

We can now decompose the expectation of $\hat{\mathcal{L}}_K^D$ as follows.

$$\mathbb{E}[\hat{\mathcal{L}}_K^D] \quad = \quad \mathbb{E} \left[ \log \frac{1}{K} \sum_{i=1}^{K} w_i \right] + \mathbb{E} \left[ \frac{\hat{\mu}_2}{2K\hat{\mu}^2} \right] \tag{60}$$

$$= \quad \log \mathbb{E}[w] - \frac{\mu_2}{2K\mu^2} + \frac{1}{2K} \left( \frac{\mu_2}{\mu^2} - \frac{1}{K} \left( \frac{2\mu_3}{\mu^3} - \frac{3\mu_2^2}{\mu^4} \right) \right) + o(K^{-1}) \tag{61}$$

$$= \quad \log \mathbb{E}[w] - \frac{1}{K^2} \left( \frac{\mu_3}{\mu^3} - \frac{3\mu_2^2}{2\mu^4} \right) + o(K^{-2}). \tag{62}$$

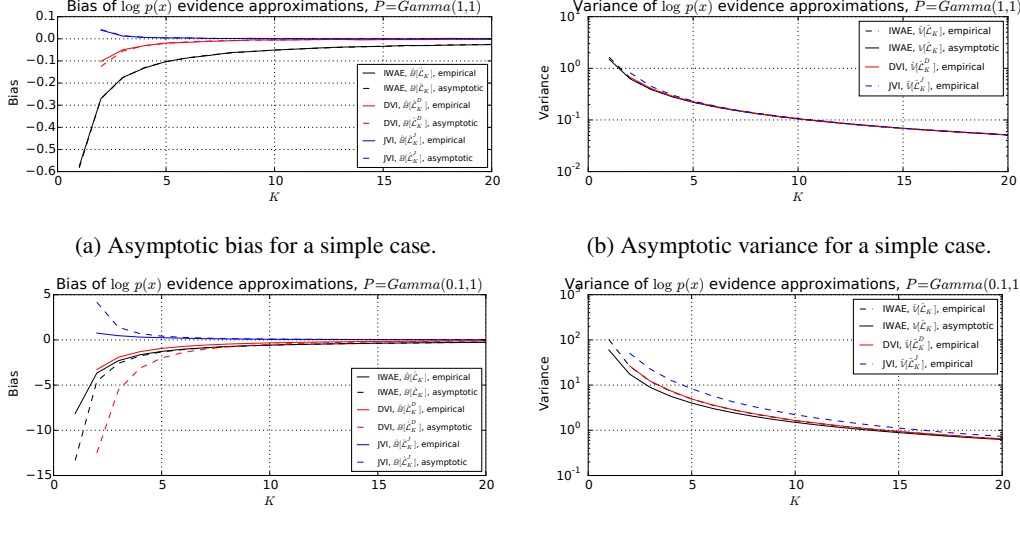

(a) Asymptotic bias for a simple case.

(b) Asymptotic variance for a simple case.

(c) Asymptotic bias for a challenging case.

(d) Asymptotic variance for a challenging case.

Figure 4: Comparing asymptotics with empirical values of bias and variance on $P = \text{Gamma}(1, 1)$ using 100,000 independent evaluations: (a)-(b) shows a simple case, $P = \text{Gamma}(1, 1)$, and (c)-(d) shows a challenging case. Observation: (a) both DVI and JVI correct for bias efficiently; (b) DVI and JVI variance closely match. (c) for a challenging case, the JVI bias is considerably smaller than the DVI bias; (d) in the challenging case, JVI has a higher variance than both DVI and IWAE.

Notably, in (62) the $1/K$ term is cancelled exactly by the delta method correction, even though we used an empirical ratio estimator $\hat{\mu}_2/\hat{\mu}^2$. Subtracting the true mean $\log p(x) = \log \mathbb{E}[w]$ from (62) yields the bias (54) and completes the proof. □

### EXPERIMENTAL COMPARISON OF DVI AND JVI

We perform the experiment shown in Figure 1 including the DVI estimator. The result is shown in Figure 4 and confirms that DVI reduces bias but that for the challenging case JVI is superior in terms of bias reduction.

## APPENDIX C: MORE JVI DETAILS

### COMPLEXITY PROOF

*Proof.* The first for loop of the algorithm has complexity $O(K)$. The second part of the algorithm considers all subsets of size $K, K - 1, \ldots, K - m$. In total these are $\mathcal{S}(K, m) = \sum_{j=0}^{m} \binom{K}{K-j} = \sum_{j=0}^{m} \binom{K}{j}$ sets. Justin Melvin derived a bound on this partial binomial sum[7], as

$$\mathcal{S}(K, m) \leq e^m \left(\frac{K}{m}\right)^m. \tag{63}$$

For each of the $\mathcal{S}(K, m)$ sets we have to perform at most $K$ operations to compute the log-sum-exp operation, which yields the stated complexity bound. □

### HIGHER-ORDER BIAS REMOVAL DEMONSTRATION

We illustrate the behaviour of the higher-order JVI estimators on the same $P = \text{Gamma}(0.1, 1)$ example we used previously. Figure 5 demonstrates the increasing order of bias removal, $O(K^{-(m+1)})$ for the $\hat{\mathcal{L}}_K^{J,m}$ estimators.

---

[7]See https://mathoverflow.net/questions/17202/sum-of-the-first-k-binomial-coefficients-for-f

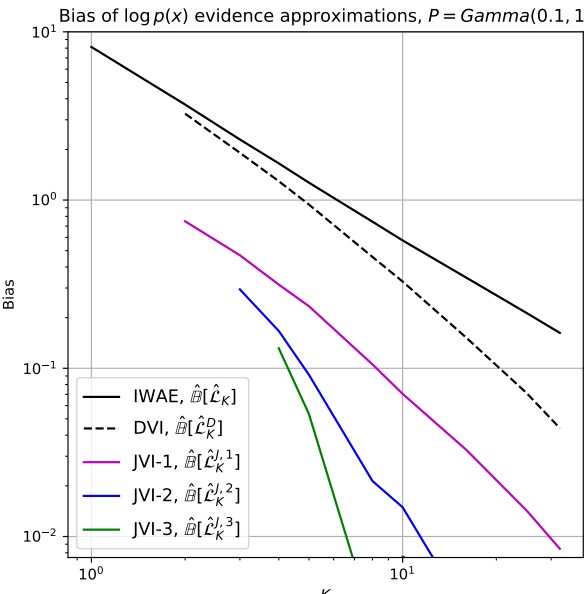

Figure 5: Absolute bias as a function of $K$.

GPU IMPLEMENTATION OF JVI

To this end let $K \geq 1$ and $m < K$ be fixed and assume the log-weights $v_i$ are concatenated in one column vector of $K$ elements. We then construct a matrix $B$ of size $(|\mathcal{S}|, K)$, where $\mathcal{S}$ is the set of all subsets that will be considered,

$$\mathcal{S} = \bigcup_{j=0}^{m} \text{EnumerateSubsets}(\{1, \ldots, K\}, K - j). \tag{64}$$

There are $|\mathcal{S}|$ rows in $B$ and each row in $B$ corresponds to a subset $S \in \mathcal{S}$ of samples so that we can use $S$ to index the rows in $B$. We set

$$B_{S,i} = \frac{1}{|S|} \mathbf{I}_{i \in S}, \tag{65}$$

where $\mathbb{I}_{\text{pred}}$ is one if the predicate is true and zero otherwise. We furthermore construct a vector $A$ with $|\mathcal{S}|$ elements. We set

$$A_S = c(K, m, K - |S|) / \binom{K}{K - |S|} = (-1)^{K - |S|} \frac{|S|! \, |S|^m}{K! \, (m - K + |S|)!}. \tag{66}$$

Using these definitions we can express the estimator as $A^\top \log(B \exp(v))$, with the $\log$ and $\exp$ operations being elementwise. However, this is not numerically robust. Instead we can compute the estimator in the log domain as $\text{logsumexp}_2(\mathbb{I}_{\mathcal{S} \times 1} v^\top + \log B) A$, where $\text{logsumexp}_2$ denotes a log-sum-exp operation along the second axis. This can be easily implemented in modern neural network frameworks and we plan to make our implementation available.

