# OpenReview forum: "Debiasing Evidence Approximations: On Importance-weighted Autoencoders and Jackknife Variational Inference"
_ICLR.cc/2018/Conference — Accept (Poster)_

### Official Review · AnonReviewer2 · 2017-11-27
**Interesting debiasing methods for Monte Carlo objectives**

**Rating:** 7
**Confidence:** 4

**Review:**

[After author feedback]
I think this is an interesting paper and recommend acceptance. My remaining main comments are described in the response to author feedback below.

[Original review]
The authors introduce jackknife variational inference (JVI), a method for debiasing Monte Carlo objectives such as the importance weighted auto-encoder. Starting by studying the bias of the IWAE bound for approximating log-marginal likelihood, the authors propose to make use of debiasing techniques to improve the approximation. For the binarized MNIST the authors show improved approximations given the same number of samples from the auxiliary distribution q(z|x).

JVI seems to be an interesting extension of, and perspective on, the IWAE bound (and other Monte Carlo objectives). Some questions and comments:

* The Cremer et al. (2017) paper contains some errors when interpreting the IWAE bound as a standard ELBO with a more flexible variational approximation distribution. For example eq. (1) in their paper does not correspond to an actual distribution, it is not properly normalized. This makes the connection in their section 2.1 unclear. I would suggest citing the following paper instead for this connection and the relation to importance sampling (IS):
Naesseth, Linderman, Ranganath, Blei, "Variational Sequential Monte Carlo", 2017.

* Regarding the analysis of the IWAE bound the paper by Rainforth et al. (2017) mentioned in the comments seems very relevant. Also, because of the strong connection between IWAE and IS detailed in the Naesseth et al. (2017) paper it is possible to make use of a standard Taylor approximation/delta methods to derive Prop. 1 and Prop. 2, see e.g. Robert & Casella, "Monte Carlo Statistical Methods" or Liu's "Monte Carlo Strategies for Scientific Computing".

* It could be worth mentioning that the JVI objective function is now no longer (I think?) a lower bound to the log-evidence.

* Could the surprising issue (IWAE-learned, JV1-evaluated being better than JV1-learned, JV1-evaluated) in Table 1 be because of different local optima?

* Also, we can easily get unbiased estimates of the evidence p(x) using IS and optimize this objective wrt to model parameters. The proposal parameters can be optimized to minimize variance, how do you think this compares to the proposed method?

Minor comments:
* p(x) -> p_\theta(x)
* In the last paragraph of section 1 it seems like you claim that the expressiveness of p_\theta(x|z) is a limitation of VAE. It was a bit unclear to me what was actually a general limitation of maximum likelihood versus the approximation based on VAEs.
* Last paragraph of section 1, "strong bound" -> "tight bound"
* Last paragraph of section 2, citation missing for DVI

---

> ### Author Response · Authors · 2017-12-15
> **Revised version of the paper available**
>
> Thank you for your comments.
>
> Regarding the errors in Cremer et al. and the connection to Naesseth et al.:
> in their latest version, https://arxiv.org/pdf/1704.02916.pdf, Section 5.2 the expected q distribution is proven to be properly normalized.
> We will add the Naesseth et al. paper as additional reference, however, the Cremer et al. work directly considers the IWAE, whereas Naesseth considers sequential Monte Carlo, so we believe the Cremer et al. paper to be the more
> appropriate reference.
>
> Relation to Rainforth et al.:
> we were not aware of this work and indeed their analysis is more general and includes our Proposition 1 and 2 as special cases.  We point this out in our revised version.
> Regarding the delta method, our proposition 1/2 are derived using the delta method for moments, as we write in the proofs to Proposition 1 and 2 in appendix A.  As reference we give Christopher Small's specialized book on the topic of asymptotic expansions in statistics (which we can recommend highly).
>
> JVI no longer a bound:
> good point, we now make this a lot clearer in Section 5 of the revised paper.
>
> JVI-trained, JVI-evaluated worse due to local minima:
> We have multiple hypotheses why this could be the case.
> 1. In line with a hypothesis that was recently put forth in another paper by Rainforth et al., http://bayesiandeeplearning.org/2017/papers/55.pdf, which is that the encoder network becomes more challenging to learn.  (You can see this by considering a perfect log-marginal likelihood objective; in that case the
> encoder gradient would vanish completely.)  To test this hypothesis we have investigated using two training objectives, where we use a regular ELBO for the encoder and a JVI or IWAE objective for the decoder.  This does lead to better encoders but initial results are not conclusively showing a benefit of JVI.
> 2. JVI estimators are no longer bounds and perhaps the optimization moves the decoder into parameters which systematically amplify positive bias.  We do not know a simple way to test this hypothesis.
>
> Importance sampling:
> using an importance-sampling evidence approximation and controlling the variance (or moment-matching) of this approximation is an interesting idea and in fact pursuing this exact idea led to the current paper.
>   The connection is as follows: for training we need log-marginal likelihoods to decompose additively over independent instances, and just as IWAE is biased, so will be importance sampling estimates (and annealed importance sampling, AIS, estimates).  For tractable sample sizes the bias is quite large, e.g. 2--4 nats for MNIST.  This made us consider debiasing corrections for IS objectives and led us to consider debiasing the IWAE.
>   Pursuing the IS route would be interesting for future work; the exact variance-optimal proposal is the true posterior p(z|x), but a sample version of minizing the variance can be used, see e.g. (Rubinstein and Kroese, "Simulation and the Monte Carlo Method", second edition, Section 5.6.2). Alternatively one can update the encoder using the regular ELBO criterion as we tried recently for our JVI objectives.
>
>
> Expressiveness of p_\theta(x|z):
> this is a general point for any latent variable model that we need to make sure that the observation model is expressive enough to model the statistics of the data.  It is not specific to VAEs.

---

> > ### Comment · AnonReviewer2 · 2017-12-20
> > **IWAE bound and IS**
> >
> > * IWAE bound
> > In your Section 2: "The bounds LK seem quite different from LE, but recently Cremer et al. (2017) and Naesseth et al. (2017) showed that an exact correspondence exists: any LK can be converted into the standard form LE by defining a modified distribution qIW(z|x) through an importance sampling construction."
> >
> > Using the modified distribution qIW(z|x) (denoted by qEW(z|x) in the Cremer et al. (2017) paper) in a standard form bound (LE in your notation) leads to a tighter ELBO than the IWAE ELBO. This is first shown in Naesseth et al. (2017), Theorem 1. Cremer et al. (2017) (updated paper) includes a special case of this result in Section 5.3, but this is attributed to the first author of Naesseth et al. (2017). It is good to also include a citation to Cremer et al. because they focus on importance sampling.
> >
> > * IS
> > I believe another main obstacle to optimize the IS-based cost functions I mention is that the variance of the stochastic gradients might be prohibitive, even if we do not need to subsample data. The reason I mentioned this procedure is that I think the motivation behind JVI, which is currently debiasing the IWAE log-marginal likelihood bound, could be further strengthened by including a brief discussion of this zero-bias version and why it isn't practical.

---

### Official Review · AnonReviewer3 · 2017-11-27
**interesting statistical analysis and ideas; experiments are limited**

**Rating:** 6
**Confidence:** 3

**Review:**

The authors analyze the bias and variance of the IWAE bound from Burda et al. (2015), and with explicit formulas up to vanishing polynomial terms and intractable moments. This leads them to derive a jacknife approach to estimate the moments as a way to debias the IWAE for finite importance weighted samples. They apply it for training and also as an evaluation method to assess the marginal likelihood at test time.

The paper is well-written and offers an interesting combination of ideas motivated from statistical analysis. Following classical results from the debiasing literature, they show a jacknife approach has reduced bias (unknown for variance). In practice, this involves an enumerated subset of calculations leading to a linear cost with respect to the number of samples which I'm inclined to agree is not too expensive.

The experiments are unfortunately limited to binarized MNIST.  Also, all benchmarks measure lower bound estimates with respect to importance samples, when it's more accurate to measure with respect to runtime. This would be far more convincing as a way to explain how that constant to the linear-time affects computation in practice.  The same would be useful to compare the estimate of the marginal likelihood over training runtime.  Also, I wasn't sure if the JVI estimator still produced a lower bound to make the comparisons. It would be useful if the authors could clarify these details.

---

> ### Author Response · Authors · 2017-12-15
> **Revised version of the paper available**
>
> Thank you for your review.
>
> Is JVI a lower bound?:
> No, it is not.  We added a clarification to the beginning of the JVI section.
>
> Evaluation with respect to runtime:
> Figure 2 shows that on a GPU, in most cases, we observe linear scaling behaviour between the number K of samples and runtime.  Therefore, any of our experiments with respect to K can be seen as also holding with respect to runtime.
> A practical issue is that runtime is more difficult to consistently assess; we did this for Figure 2 on a single-user GPU workstation, but for the other experiments, in our multi-user GPU cluster system we cannot ensure consistent timings over many training runs so instead we report the number of samples.

---

### Official Review · AnonReviewer1 · 2017-11-27
**Interesting analysis**

**Rating:** 7
**Confidence:** 3

**Review:**

This paper provides an interesting analysis of the importance sampled estimate of the LL bound and proposes to use Jackknife to correct for the bias. The experiments show that the proposed method works for model evaluation and that computing the correction is archivable at a reasonable computational cost. It also contains an insightful analysis.

---

### Public Comment · (anonymous) · 2017-10-31
**Similar results on IWAE bound missing from literature review**

Hey

I just wanted to draw your attention to the recent preprint https://arxiv.org/pdf/1709.06181.pdf which includes a very similar set of results to your analysis of the IWAE bound in section 3.  They consider convergence bounds for a more general class of problems by using bias-variance decompositions and consider the IWAE (in section 6.5) as a particular example which leads to a result which is effectively equivalent to your Propositions 1 and 2 up to a constant factor.  To see this, note that your result in corresponds to the case of N=1, M=K, and that the C^2_0 ς^4_1 / 4M^2 + O(1/M^3) terms in their bound constitute the biased squared with the variance comprising of all the other terms in their bound (see proof of Theorem 3).  They thus show the same key high-level result that the bias and variance are both O(1/K).

Given how recent this related work is and that it is only a preprint with a predominantly different focus, I don't think this detracts too much from your current submission, but you may wish to revise the paper to acknowledge that this very similar result was previously independently derived and to highlight the differences of your results from theirs.

---

> ### Author Response · Authors · 2017-11-01
> **Thanks**
>
> Thank you for pointing to this related work, it seems clearly relevant; we will read it and will add a citation and potentially a more detailed discussion of the relation to the next version of our submission.

---

> ### Author Response · Authors · 2017-12-15
> **Revised version of the paper**
>
> We now read the Rainforth et al. paper and agree with your assessment.
> We added an appropriate discussion before our analysis in Section 3 of the
> revised submission.

---

### Comment · AnonReviewer1 · 2017-11-27
**Interesting analysis**

Hello,

Interesting analysis. But I’m not particularly surprised that JVI during training does not result in better models compared to IWAE. We often observe very small effective sampling sizes when training big models with only few of the normalized importance weights is close to 1. It seems JVI would result in very similar gradients under these circumstances. I think it could strengthen the paper if this would be investigated and discussed.

Wouldn’t a common scale for the LL plots in Figure 3 make it easier to read?

Do you think it would be preferable if the community continued to report biased bounds instead of JVI estimated LLs? This provides a natural partial protection against overestimating due to low effective sampling sizes, doesn’t it?

---

> ### Author Response · Authors · 2017-12-15
> **Revised version of the paper available**
>
> Thank you for your review.
>
> Small effective sample size:
> Indeed the effective sample size (ESS) for IWAE can be very small because the log-weight distribution has high variance. The intuition that this would lead to very similar gradients is wrong however: consider JVI-1, which is a weighted average of leave-one-out IWAE objectives.  If the ESS is close to one, then one of the leave-one-out IWAE objectives will not contain the dominant sample but be the IWAE on the remaining samples.  The JVI gradients are weighted averages of IWAE gradients and will effectively downweight dominant samples.  JVI-2 and higher-order variants have an even stronger effect because they leave out more than one sample.
>
> Shared scale in Figure 3:
> We agree this could be useful for comparing results across training objectives; the main purpose of Figure 3 is to demonstrate that within any regime higher order JVI estimates reduce bias and because the scales are quite a bit different between training objectives we opted to use space efficiently to that end.
>
> Reporting bounds versus debiased LL estimates:
> We do not have a comprehensive answer to this point and both views have merit:
> 1. Reporting bounds across models provides an estimate of the model performance that is conservative/safe against deficiencies in tuning the inference procedure.
> 2. Reporting more accurate LL estimates has the advantage of being a more accurate assessment of the model performance free of model-specific biases; for example, an ELBO may be bad in case the encoder is bad, despite the
> generative model being of good quality.  Also, in many cases the LL directly transfers into a natural metrics such as bits-per-pixel needed for compression.

---

### Author Response · Authors · 2018-04-24
**Talk recording available**

For anyone interested, there is a talk recording available related to this work:

https://www.youtube.com/watch?v=nRgjvACKNAQ

---

### Decision · Program_Chairs · 2018-01-29
**ICLR 2018 Conference Acceptance Decision**

**Decision:**

Accept (Poster)

**Comment:**

The authors analyze the IWAE bound as an estimator of the marginal log-likelihood and show how to reduce its bias by using the jackknife. They then evaluate the effect of using the resulting estimator (JVI) for training and evaluating VAEs on MNIST. This is an interesting and well written paper. It could be improved by including a convincing explanation of the relatively poor performance of the JVI-trained, JVI-evaluated models.